# Evaluation of Blood Pressure Status and Mortality in Turkey: Findings from Chronic Diseases and Risk Factors Cohort Study

**DOI:** 10.3390/medicina59081366

**Published:** 2023-07-26

**Authors:** Kaan Sozmen, Gul Ergor, Sibel Sakarya, Gonul Dinc Horasan, Ceyda Sahan, Banu Ekinci, Ahmet Arikan, Secil Sis, Belgin Unal

**Affiliations:** 1Department of Public Health, Faculty of Medicine, Izmir Katip Celebi University, Izmir 35620, Türkiye; 2Department of Global Health and Population, Harvard T.H. Chan School of Public Health, Boston, MA 02115, USA; belgin.unal@deu.edu.tr; 3Department of Public Health, Faculty of Medicine, Dokuz Eylul University, Izmir 35340, Türkiye; gul.ergor@deu.edu.tr (G.E.); ceyda_sahan@hotmail.com (C.S.); 4Department of Public Health, Faculty of Medicine, Koc University, Istanbul 34010, Türkiye; 5Department of Public Health, Faculty of Medicine, Izmir University of Economics, Izmir 35330, Türkiye; dincgonul@gmail.com; 6Department of Chronic Diseases and Elderly Health, General Directorate of Public Health of Turkey, Ankara 06430, Türkiye; drbanutek@yahoo.com (B.E.); ahmetarikan06@gmail.com (A.A.); secil.sis@saglik.gov.tr (S.S.)

**Keywords:** arterial hypertension, cohort study, survival, cardiovascular disease

## Abstract

*Background and objectives*: An important Non-Communicable Disease risk factor, hypertension (HT), is highly prevalent and controlled HT rates are not sufficient which increases the risk of developing premature deaths. The purpose of the study is to evaluate differences in all-cause and cardiovascular-related mortality according to HT status by using national data from Chronic Diseases and Risk Factors Survey in Turkey (2011–2017). *Materials and Methods:* Cox regression models were used to estimate hazard ratios (HR) for predicting the all-cause and cardiovascular system-related mortalities. Median follow-up period was 6.2 years. *Results:* Among individuals with HT, 41.8% was untreated, 30.1% received treatment and had controlled blood pressure, and 28.1% were under treatment but had uncontrolled BP levels. The hazard for mortality among treated & uncontrolled hypertensive participants was significantly higher for all-cause (HR = 1.32, 95% CI = 1.06–1.65), cardiovascular (HR = 2.11, 95% CI = 1.46–3.06), heart disease (HR = 2.24, 95% CI = 1.46–3.43), and Coronary Heart Disease mortality (HR = 2.66, 95% CI = 1.56–4.53) compared to normotensive participants. *Conclusions:* Individuals with HT who were treated but do not have controlled blood pressure in Turkey had a significantly increased risk of Cardiovascular Disease and all-cause mortality. Along with studies investigating the causes of uncontrolled blood pressure despite initiation of treatment, support should be provided to patients in cases of non-adherence to antihypertensive medication or life change recommendations.

## 1. Introduction

In the year 2019, 17 million premature deaths occurred due to non-communicable diseases (NCDs) [1]. Among these premature NCD deaths, 38% were caused by CVDs and more than three-quarters of CVD deaths occurred in developing countries [1].

High blood pressure, often known as hypertension (HT), is associated with an increased risk of coronary heart disease (CHD), heart failure, cerebrovascular illness, and premature death. Due of its significant global burden, HT is regarded as one of the major public health issues [2,3]. The main underlying risk factors for CVD events were metabolic variables, according to a prospective study involving 155,722 participants from developing countries; HT had the largest percentage of population-attributable risk (22.3%) [4].

Sustainable global targets for NCDs is to reduce the premature deaths by one-third and the prevalence of HT by 25% between 2010 and 2030 [5]. For example, a 10-mmHg decrease in systolic blood pressure was linked to a 20% decreased risk of experiencing cardiovascular disease events, highlighting the significance of treating high blood pressure in preventing cardiovascular morbidity [6,7]. However, global awareness rates for HT were 59% among women and 49% among men in 2019. It was reported that 47% of women and 38% of men were under medical treatment and in total 21% had controlled blood pressure [8]. In Turkey, NCDs are estimated to account for 86% of total deaths, and approximately 20% of adults die prematurely. According Ministry of Health (MoH) Statistical Yearbook-2019; in total, 435,941 deaths occurred and circulatory system diseases (ICD code: I00-I99) were responsible for 33.8% of deaths in males and 40.3% of deaths in females and 37% in total [9].

Turkey continues to have a high CVD mortality rate compared to the European region despite decreasing trends in CVD mortality reported over the last two decades. [10,11]. HT is highly prevalent among the adult population; approximately 32.7% of individuals over age 20 have HT in Turkey [12]. According to an epidemiological modeling study which explains the contributions of each CVD risk factor to the decline in CHD mortality rates from 1995 to 2008 in Turkey, the mean blood pressure decline avoided or postponed the largest contribution to CHD deaths by accounting for 29% of the total reduction in CHD mortality [10]. Even a declining trend in mean blood pressure levels in Turkey was observed during the last decades awareness, adherence to treatment and control rates for HT remain low [13].

People with unmanaged HT have higher risk for mortality. The majority of the risk comparisons for mortality for individuals with HT who receive treatment and have controlled BP, treated but have uncontrolled BP levels, and untreated, when compared to normotensives, comes from clinical trials or cohort studies conducted in developed countries. The epidemiological data from developing countries is not sufficient and there are a limited number of cohort studies that can provide data for detailed analysis.

This study aims to evaluate the differences in all-cause and cardiovascular-related mortality according to hypertension status by using national data from Turkey.

## 2. Materials and Methods

### 2.1. Study Population and Data Collection

The study data were obtained from the Chronic Diseases and Risk Factors Survey (CDRFS), 2011, and CDRFS Cohort Study-2017 conducted by the MoH of Turkey. Survey methodology and summary findings have been presented earlier [14,15,16]. The survey has a population-based closed cohort design. The minimum sample size was determined as 16,622 in order to detect 1% prevalence with a 0.15% deviation for the survey. The study population was formed by individuals who were aged over 15 years and registered within the Family Physician System in Turkey. Participants were randomly sampled from each family medicine unit that provides primary care services. Family physicians (FPs) gathered the anthropometric data and blood pressure readings.

Venous blood samples were obtained in Family Health Centers (FHC) and samples were analyzed in Public Health Laboratories that operate in the framework of the Ministry of Health. In total, 18,477 individuals completed the questionnaire and 80.5% of them (n = 14,887) had their blood sampled by visiting FPs. We analyzed the data from 2011 to 2017, including 14,337 participants without missing information for blood pressure and laboratory measurements. The flow diagram of the cohort is presented in Figure 1.

The ethical review board of the Turkish MoH gave its approval to this study. Individuals were informed about the study and that the data will be confidential. The survey questions were administered face-to-face, and data was input electronically at primary health care centers after obtaining participants’ written consent.

### 2.2. Outcomes

The main outcomes of this study are mortality due to Coronary Heart Disease, Cardiovascular Disease, Cerebrovascular Disease, and all-cause mortality. The mortality data were abstracted from the electronic Death Notification System (DNS) of the Ministry of Health. The TURKSTAT Death Certificate, which also has a section for reporting the cause of death, is completed by the family physician when a death occurs in hospital. In metropolitan areas, municipalities or primary care physicians fill out the death certificate for a death that happens at home. The 10th version of the International Classification of Diseases (ICD-10) was used by doctors to code the cause of death, and information was then entered into the DNS. Following this reporting, the deceased may be formally buried.

The cause of death recorded by the standardized method is transferred in connection with the Turkish Identity number to the Family Physician Information System and Social Security databases by electronic means. Data quality and coverage of the death notification system in Turkey have been improved reasonably by the revision in 2009 [17]. 

In this study, deaths were classified as all-cause, Coronary Heart Disease (I20-I25), heart disease deaths (including CHD and heart failure(I50)), and cardiovascular disease (CHD and stroke (I60-69)). The mortality data were obtained between June 2011 and 31 December 2017. The duration of follow-up was determined for each individual separately based on the information regarding the date of the interview and observation period of the study or the date of death.

### 2.3. Covariates

The independent variables included sociodemographic properties, lifestyle factors, clinical variables, and anthropometric measurements. The participants’ ages were divided into three groups: 18–39 years, 40–59 years, and over 60 years. Marital status was categorized as married and single/divorced/widowed. The highest level of education was categorized as: elementary school or lower, high school, and university degree. According to the Turkish Statistical Institution’s (TURKSTAT) classification, settlements with a population of at least 20,000 people were classified as urban, while those with a population below 20,000 people were accepted as rural areas [18].

Smoking status was categorized into regular smokers (individuals who currently smoked at least one cigarette a day), ex-smokers, and non-smokers. Individuals were considered as having healthy nutrition if they consume at least five portions of fruit and vegetable a day [19]. Individuals were considered as being physically active if they exercise at least 5 days per week for a duration of minimum 30 min of moderate-intensity activity such as gardening or fast walking, or vigorous-intensity aerobic activities such as cycling, running/jogging, swimming, playing soccer on at least three days per week for at least 20 min [20]. A graded (0–100) Visual Analog Scale (VAS) was used to measure perceived health status in order to evaluate health-related quality of life, with 0 points denoting “the worst health you can imagine” and 100 points denoting “the best health you can imagine”. Abdominal obesity was assessed with waist circumference measurement, and it was categorized as high if it is ≥102 cm in men and ≥88 cm in women. Any study participant with a body mass index (BMI) ≥30 kg/m^2^ was considered to have obesity [21].

Chronic diseases are assessed based on the self-reporting of participants and the history of treatments made by FPs and lab tests. CHD was evaluated by asking individuals whether they were diagnosed with acute myocardial infarction, angina pectoris, or received interventions such as angioplasty or coronary artery bypass surgery. Stroke was evaluated by asking “Have you ever been diagnosed with stroke by a physician? (yes/no)”. Multimorbidity was defined as co-occurrence of at least two chronic conditions or diseases out of 11 conditions/diseases assessed in the study [22].

Nurses at FHC collected the venous blood sample after the patient had been fasting for at least 8 h in order to measure the patient’s fasting plasma glucose (FPG) and assess their total cholesterol, LDL cholesterol, HDL cholesterol, and triglyceride. LDL-C was calculated using the Friedewald formula. FPG was evaluated with the glucose oxidase method. The participants who stated a history of diabetes based on physician diagnosis were further assessed for chronic disease reports and the medications used. Type 2 diabetes was defined as having a fasting blood sugar level greater than or equal to 126 mg/dL or by taking antidiabetic medication [23].

Instructions for blood pressure measurement have been given to family physicians. Based on measurements of Systolic Blood Pressure (SBP), Diastolic Blood Pressure (DBP), and the use of hypertension medications, individuals’ blood pressure levels were categorized into four separate groups [24].

Individuals were defined as “Normotensive” if they had SBP < 140 mmHg, DBP < 90 mmHg, were not diagnosed with hypertension, and not using medication due to high blood pressure [25]. Individuals were considered as “Not Treated” if they had a blood pressure ≥ 140/90 mmHg and were not diagnosed by a physician or do not receive any pharmacologic treatment for HT.

Hypertension was considered to be controlled if an individual was receiving treatment for high blood pressure at the time of the interview who had SBP < 140 mmHg and DBP < 90 mmHg. Uncontrolled HT was defined as having SBP ≥ 140 mmHg or DBP ≥ 90 mmHg in an individual with a history of hypertension despite receiving medication for high blood pressure. Framingham risk score (FRS) was used to estimate the 10-year risk for developing cardiovascular disease and individuals were considered as having a high risk if their FRS was 20% or more [26].

### 2.4. Statistical Methods

In descriptive statistics, continuous variables were presented as mean ± standard deviation (SD). Categorical variables were presented with percentages. The differences between blood pressure categories were assessed using the Analysis of Variance (ANOVA) for continuous variables and the chi-square test for categorical data.

All-cause, CVD, CHD, and heart disease mortality rates were calculated for each category of hypertension status, and they were presented as per 100,000 person-years with their 95% Confidence Intervals (CI). For each mortality outcome, incidence rate ratios (IRR) were calculated using Poisson regression models. In the survival analyses, the Kaplan–Meier survival curves were generated to visually illustrate the differences in the probability of mortality over time between each blood pressure group for each of the mortality outcomes. Log-rank tests were used to compare the differences in survival times between HT groups.

Two Cox proportional hazard regression models were generated to estimate Hazard Ratios (HRs) which show the independent association between HT categories and outcomes; the baseline model was adjusted for age and sex, and the second model was a multivariable model that takes into account confounding factors (age, gender, education level, area lived(rural), smoking status, fruit and vegetable consumption, physical activity, CHD history, stroke history, DM, BMI, Total Cholesterol, and VAS). The variables were considered potential confounders if these variables had *p* < 0.10 at univariable analysis.

The nonlinear relationship between blood pressure as a continuous variable and mortalities was examined using a Cox regression model with restricted cubic spline (RCS). In this study, an RCS regression line with four knots located at the 5th, 35th, 65th, and 95th percentiles was used. The number of knots for decomposing blood pressure point was confirmed based on the Akaike information criterion (AIC). The median values for blood pressure levels were used as the reference point in the analyses. The Wald test was used to examine whether there was a significant deviation from linearity in the statistical relationship between blood pressure levels and mortality. The spline curves were constructed using the ‘xbrcspline’ and ‘mkspline’ commands in STATA.

Survey weights were generated in order to correct for different nonresponse rates of provinces. The statistical analyses were performed by STATA software (Version 15, TX, USA) and Statistical Package for Social Sciences SPSS (version 22, Armonk, NY, USA). A two-sided *p* < 0.05 was considered statistically significant.

## 3. Results

### 3.1. Baseline Characteristics

In total, data from 14,437 individuals were analyzed. Out of these participants, 53.1% of them were female. The mean age of women and men was 41.9 ± 17.3 and 41.54 ± 17.2 years, respectively (*p* = 0.188). Approximately 26.5% of the individuals had HT. Among individuals with HT, 41.8% was untreated, 30.1% received treatment and had controlled blood pressure (BP), and 28.1% were under treatment but had uncontrolled BP levels. The treatment uptake rates by gender among individuals with HT were 42.7% for males and 59.3% for females. Among individuals with HT, 25.7% of the males and 33.3% of females had controlled BP.

Of the individuals with HT, 57.8% had mild, 9.5% had moderate, and 2.6% had severe hypertension. The prevalence of mild, moderate, and severe HT among untreated individuals was 85.4%, 10.6%, and 4.0%, respectively. The mild, moderate, and severe HT prevalence was 78.6%, 18.1, and 3.3% among individuals with uncontrolled BP, respectively. In total, 26.5% (n = 940) of the individuals with HT had isolated systolic hypertension. The prevalence of isolated systolic hypertension among untreated and uncontrolled BP was 46.0% and 25.9%. The prevalence of isolated diastolic hypertension was 9.3% (n = 333) among individuals with HT. The prevalence of isolated diastolic hypertension among untreated and uncontrolled BP was 16.7% and 11.3%, respectively.

The baseline characteristics of the participants stratified by HT status are presented in Table 1. The distribution of independent variables showed significant differences by hypertension status (*p* < 0.001). When compared to other BP categories, those with uncontrolled BP had a significantly larger percentage of people with the lowest level of education (primary school or less) (*p* < 0.001). The proportion of obese individuals determined by BMI and waist circumference was higher in the uncontrolled BP group compared to their respective BP groups. Individuals in any of the high blood pressure groups had significantly higher mean total cholesterol and LDL cholesterol levels compared to ones with normal BP levels. People in the treated and uncontrolled BP groups had higher rates of CHD, DM, stroke, and multimorbidity. People with uncontrolled blood pressure were more likely to be older, female, and to reside in rural areas. These people also had lower VAS ratings.

### 3.2. Survival

In total, 685 deaths occurred over the course of a median follow-up period of 6.2 years. Cardiovascular deaths formed approximately one-third of total deaths (n = 225). The 6-year survival rates for participants with normal BP, untreated hypertension, treated & controlled, and treated & uncontrolled HT were 97.6%, 91.9%, 87.2%, and 83.6%, respectively. The Kaplan–Meier Survival Curves depicting differences in survival probabilities by BP status are presented in Figure 2.

The log-rank tests comparing the survival distributions presented significant differences between all high blood pressure groups for overall survival, cardiovascular mortality, and heart disease mortality. Individuals with uncontrolled BP had a significantly higher cumulative probability of mortality compared to other blood pressure categories (log-rank *p* < 0.001). Individuals with controlled HT and untreated HT had a lower cumulative probability of CHD mortality compared to individuals with uncontrolled HT.

The differences in mortality for CHD between controlled HT and untreated HT groups did not differ significantly (*p* = 0.3415).

All-cause, cardiovascular, heart disease, and CHD mortality rates were 650.15 (95% CI: 603.73–701.12), 213.55 (95% CI: 187.79–243.92), 156.61 (95% CI: 134.81–183.02), and 104.40 (95% CI: 86.94–126.52) per 100,000 person-years, respectively. Table 2 presents the mortality rates, incidence rate ratios (IRRs), and Hazard Ratios stratified by hypertension categories. The IRRs were significantly higher for the treated & uncontrolled group. In the baseline model, which was adjusted for age and gender, the HRs for total mortality, CVD mortality, CHD mortality, and heart disease mortality were significantly higher for both treated & uncontrolled and treated & controlled groups compared to normotensives.

According to fully adjusted cox regression model using all potential confounders, the hazard for mortality among treated & uncontrolled hypertensive participants was significantly higher for all-cause (HR = 1.32, 95% CI: 1.06–1.65), cardiovascular (HR = 2.11, 95% CI: 1.46–3.06), heart disease (HR = 2.24, 95% CI: 1.46–3.43), and CHD mortality (HR = 2.66, 95% CI: 1.56–4.53) compared to normotensive participants.

Association of SBP and DBP as a continuous variable with Hazard ratios for mortalities are presented in Figure 3 and Figure 4. When SBP and DBP were used as continuous variables in restricted cubic spline analysis, the natural cubic spline curve demonstrated a positive nonlinear monotonic relationship with mortalities. The hazard ratios for mortality became larger when systolic blood pressure increased over 120 mmHg.

We used median values of BP as inflection points; 120.0 mmHg for SBP and 80.0 mmHg for DBP were determined as having the lowest hazard ratio for mortalities. The association between blood pressure and risk of mortality was assessed by performing two separate (piecewise) Cox Regression analyses on stratified data based on blood pressure inflection points (Table 3). Systolic Blood Pressure values higher than 120 mmHg was associated with increased risk of all-cause mortality, suggesting that for every increase of 10 mmHg in SBP higher than the reference value of 120 mmHg was associated with a 7% increased risk in all-cause mortality (HR = 1.07, 95% CI: 1.02–1.12). The risk of coronary artery disease-related deaths increased by 46% (HR = 1.46, 95% CI: 1.17–1.84) for each 10-mmHg increment increase in DBP over 80 mmHg.

We present HRs regarding covariates stratified by BP groups in Figure 5. The associations of the majority of the independent variables with mortalities were insignificant according to multivariable models. DM was associated with CHD and all-cause mortalities among individuals with controlled BP. Among the individuals with controlled BP, the hazard for mortality in diabetics was significantly higher for all-cause (HR = 1.75, 95% CI = 1.22–2.49) and Coronary Heart Disease mortality (HR = 3.44, 95% CI = 1.29–9.14) compared to individuals without DM.

## 4. Discussion

In the study, approximately one-third of the individuals had HT. Among individuals with HT, one-third had controlled BP levels and approximately 28% had uncontrolled BP. Even though treatment rates were higher among women, they were also more likely to have uncontrolled blood pressure. Treatment uptake rates and control rates increased with age. These findings are in concordance with previous reports from Turkey. According to the National Household Health Survey in Turkey-2017 (n = 6053), which used the Stepwise approach of WHO, 27.7% of the participants had a history of high blood pressure or were currently receiving medication for HT. Among those previously diagnosed with HT, 72.7% were currently on treatment for HT. In the same study, treatment uptake rates increased by higher age; it was lowest in the group aged 15–29 (29.6%) and it was highest among individuals aged ≥70 (85.4%). The BP was under control for 23.8% of the individuals with HT (18.5% for men and 28.4% for women). The proportion of controlled blood pressure among individuals with HT increased gradually from 7.6% in the 15–19 age group to 30.0% at age 70 and over in both sexes [27]. According to a recent study from Iran with 163,770 participants, aged 35–70 years, the prevalence of HT was 22.3%. In the same study, 77.5% of the participants were aware, 82.6% of individuals with HT received treatment, and 63.7% of them had controlled BP. Treatment uptake rates increased by age and were higher among females. The percentage of patients with controlled blood pressure was 75.9%, while control rates declined with age and were lower in women [28].

We found that individuals who were treated but had uncontrolled BP had the highest risk for all-cause, CVD, CHD, and heart disease mortality than normotensive adults. According to previous research, it is evident that especially higher SBP was associated with increased risk for end organ damage, mortality, and higher overall CHD risk scores which can explain this finding [7]. According to the restricted cubic spline regression, analysis curves showed a nonlinear relationship between mortalities and blood pressure levels. Especially, higher SBP values resulted in increased risk for mortality and lower values for SBP were associated with increased risk for heart disease-related mortalities. This finding is in concordance with previous research indicating that both very high and very low SBP values might have a negative impact on survival rates [29,30].

Low adherence to antihypertensive drugs may be one of the causes of uncontrolled blood pressure in the general population. Low health literacy levels, a lack of a social or familial support system, and a lack of health insurance are just a few of the characteristics that are linked to low adherence [31,32]. An earlier study from Turkey that examined the connection between blood pressure control and health literacy found that older people and those with a lower level of education were more likely to have low health literacy levels [33]. It was also found that individuals with lower health literacy levels had higher rates of uncontrolled BP. It was reported that health literacy is related to cognitive skills together with social skills like finding, understanding, and interpreting knowledge [34]. In our study, we found that the individuals with uncontrolled BP had lower education levels and were older compared to other BP groups which might have a negative impact on BP control due to low health literacy levels.

In the study, individuals with uncontrolled BP reported higher rates of history of Coronary Artery Disease, Stroke, and DM. Previous research suggests that BP control threshold levels are lower if individuals have DM compared to non-diabetics [35,36]. Comorbidity of HT and DM is common and it could be because both HT and DM have common factors involved in the pathogenesis of both conditions, such as increased levels of renin-angiotensin-aldosterone causing higher sodium reabsorption in the kidneys and stimulated sympathetic nervous system, which have an impact on vascular tonus, higher exposure to oxidative stress, and inflammation [37,38,39]. In our study, the obesity rates were higher among individuals who were treated & uncontrolled BP. According to previous research, BMI is positively correlated with higher BP values [40]. A recent study found that; obese people’s blood pressure remained higher despite receiving more antihypertensive medication than leaner individuals, suggesting that obese people would require higher dosages to drop their blood pressure to the same extent as leaner people [41]. One of the possible explanations is obesity has a negative impact on cardiac output due to intravascular volume expansion and reduction in cardiac contractility [42].

When we adjusted regression models for only age and sex, we found that treated & controlled hypertensives had a considerably higher risk of mortality from all causes and CVD than normotensives. However, after considering several other covariates in the multivariable analysis, the mortality risk among individuals who were treated and controlled did not differ significantly compared to normotensives.

Previous population-based prospective studies from Finland and the USA revealed that the risk of all-cause and CVD mortality was not statistically different between normotensives and hypertensives with regulated blood pressure [43,44]. According to a study from the United States using the Third National Health and Nutrition Examination Survey data with 13,947 participants aged over 18, the mortality rates did not differ significantly between normotensives and individuals who were treated and had controlled blood pressure [45]. However, research from the United Kingdom and Germany which was conducted among older individuals reported that intense control of BP resulted in higher mortality rates [46]. However, a study using information from the Multi-Ethnic Study of Atherosclerosis (MESA) and the Coronary Artery Risk Development in Young Adults (CARDIA) study found that people with treated and controlled hypertension had a higher mortality risk than normotensives. That research stated one of the possible explanations as: the participants with controlled blood pressure who are on antihypertensive medication experience significantly higher cumulative BP exposure over time than untreated people. In our research, individuals who were treated and kept under control had higher age- and sex-adjusted HRs for all-cause mortality, cardiovascular mortality, and heart disease mortality compared to normotensives.

In our study, the mortality rates were lower among individuals who were untreated compared to ones who received treatment irrespective of BP control. However, some studies showed higher mortality rates among untreated compared to treated & controlled. The main reason for this finding in our study could be due to the low overall CHD risk determined by FRS. FRS equation consists of main CHD risk factors such as age, gender, total cholesterol (TC), high-density lipoprotein cholesterol (HDL), smoking status, SBP, DBP, and presence of DM [26].

In our study, the individuals who had HT but were untreated were younger compared to other groups, which might explain lower mortality rates in this group. Another reason could be the lower burden of other major cardiovascular risk factors as these individuals had lower mean cholesterol levels and they had lower prevalence for DM. Other studies from developing countries had an older population and higher overall risk profile compared to our data. In our study, individuals who were untreated also had lower rates for the history of CHD and Stroke and lower rates for the presence of multimorbidity. The short duration of follow-up in the study, a lower duration of hypertension in untreated hypertensive subjects due to young age, and a higher prevalence of mild hypertension in untreated hypertensive individuals might be other possible explanations. Even though the proportion of individuals with high FRS among untreated was lower compared to other BP groups, some of the unhealthy lifestyle factors were more prevalent among these individuals, such as higher smoking rates and lower fruit and vegetable consumption compared to individuals who were treated. American Heart Association (AHA) guidelines recommend lifestyle modification for 3–6 months in the first place to individuals who have a low risk for developing CHD and if BP targets cannot be achieved, then treatment is recommended to these individuals [47]. The primordial and primary prevention efforts targeting all populations should be implemented such as increasing levels of physical activity, reducing smoking, and limiting salt consumption. Another strategy to be implemented in primary care settings could be the routine screening of adults aged over 18 years by yearly measurements of BP by FPs, as recommended in Family Medicine Periodic Health Inspections Guideline-2015 in Turkey [48]. According to National Household Health Survey in Turkey, 13.6% of respondents had never had their blood pressure levels measured [27]. Routine screening for HT will be able to identify undiagnosed people and improve adherence to treatment in those who have previously received a diagnosis. To keep their blood pressure under control, hypertensive adults need to be treated and monitored.

There are several action plans and programs prepared by MoH of Turkey that are targeting the control of the CVDs and their risk factors: the Turkish Cardiovascular Diseases Prevention and Control Program (2021–2026), the Program for Reducing Excessive Salt Consumption (2017–2021), National Tobacco Program (2018–2023), Turkish Diabetes Program (2015–2020), Healthy Nutrition and Active Life Program (2018–2023). Public Health-related goals for the CVD control program are increasing awareness of cardiovascular diseases and risk factors in society, improving physical activity habits for individuals, shaping the dietary habit of the society into healthy ones, calculating the risk of developing cardiovascular diseases, and planning interventions in accordance with the risk score [49].

Our study has important strengths to consider. Firstly, this is the first cohort study planned at the national level with a large number of participants, aimed to determine the incidence of NCDs in Turkey. For the first time, record linkage of electronic health data (Family Medicine Information System, Social Security Institute, and e-Pulse) through Turkish Identity number was used. Second, the findings during the analyses of the data findings were adjusted for several covariates including sociodemographic, clinical, and lifestyle factors.

Our study has various limitations that should be mentioned. Firstly, in the analysis, BP data came from baseline measurement. The blood pressure values might change over time which has an impact on the magnitude of the estimates. The treatment uptake rates, antihypertensive drug type, dosage of drugs, and adherence to medications might change during the observation period and information regarding this could help further evaluation of individuals who were receiving treatment and had controlled BP. Although the findings were adjusted by using multivariable models, there could be some unmeasured confounding due to lack of detailed data on dietary habits and some lifestyle factors which might have an influence on BP, such as salt consumption. The duration of HT was not evaluated; however, the age of diagnosis and duration of HT might have an impact on mortality. Another limitation is that we used the FRS equation to estimate 10-year risk for developing CVD which might result in under or overestimation of the risk score values. According to the TEKHARF cohort conducted in Turkey, the Framingham risk score underestimated the CVD risk [50]. Another study from Turkey found both FRS and SCORE models were reliable for detecting the presence and severity of CVDs [51]. Further research is needed to generate risk score equations specific to Turkish settings. In our study, only mortality were evaluated; however, the occurrence of morbid cardiovascular events may vary by HT status, which can be considered in future research.

## 5. Conclusions

This study suggests that individuals with HT who were treated but do not have controlled blood pressure in Turkey have a significantly increased risk of CHD, Cardiovascular Disease, CVD, and all-cause mortality. The mortality risk is also higher among individuals who were treated and had controlled blood pressure but the difference was not statistically significant. Due to the low CHD risk profile among the individuals who did not receive treatment, the mortality rates were similar to normotensives. Our study highlights the need for a life course approach for all individuals starting with primary prevention efforts targeting healthy individuals by promoting healthy lifestyle factors. Due to low awareness rates for HT and inappropriate adherence to medications, implementation of secondary and tertiary prevention strategies, scaling-up screening rates and provision of treatment at the early stage, increasing health literacy, and reminding the family physicians for screening their patients with HT to manage blood pressure control status would be beneficial for reducing mortality rates, especially by effective control of blood pressure in Turkey. Further studies are required to better understand the contribution of non-adherence to medications and resistant hypertension to uncontrolled blood pressure in local settings.

## Figures and Tables

**Figure 1 medicina-59-01366-f001:**
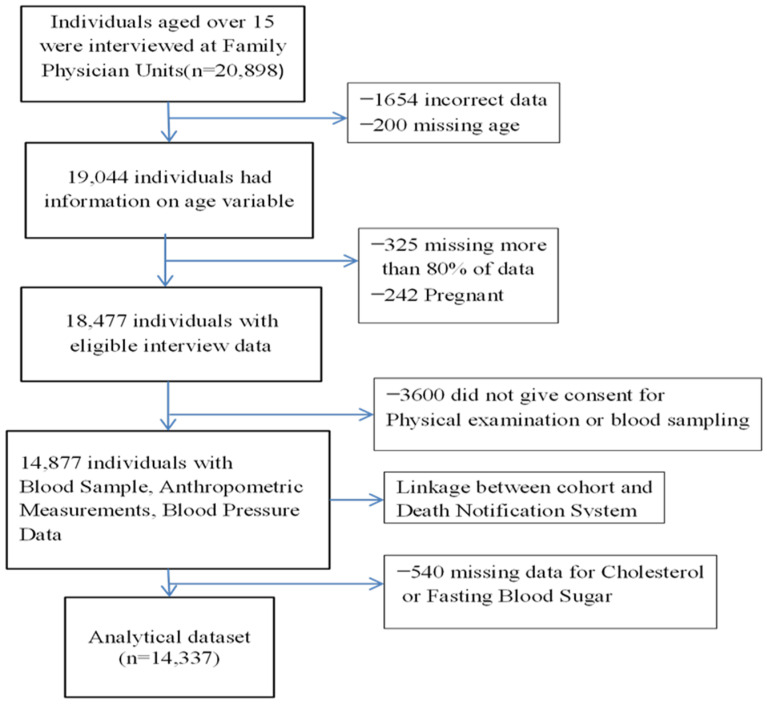
Flow chart of the cohort study.

**Figure 2 medicina-59-01366-f002:**
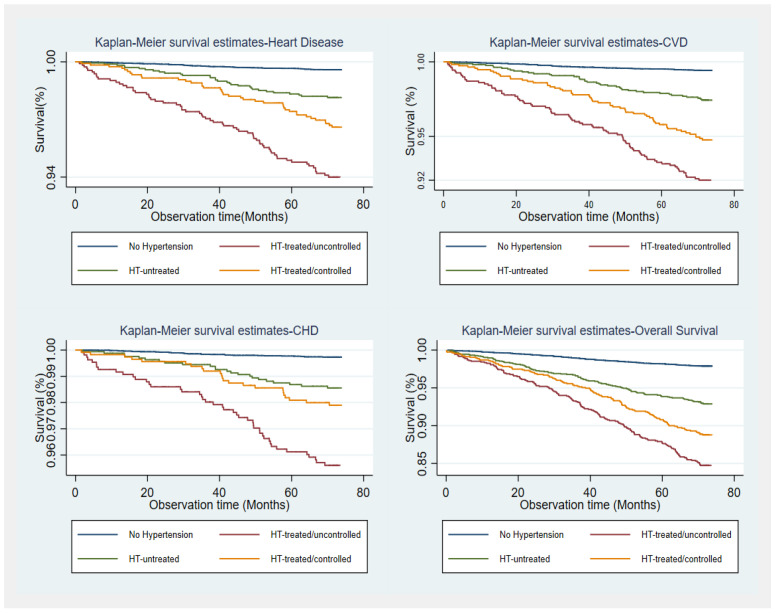
Kaplan–Meier Survival Curves stratified by blood pressure status.

**Figure 3 medicina-59-01366-f003:**
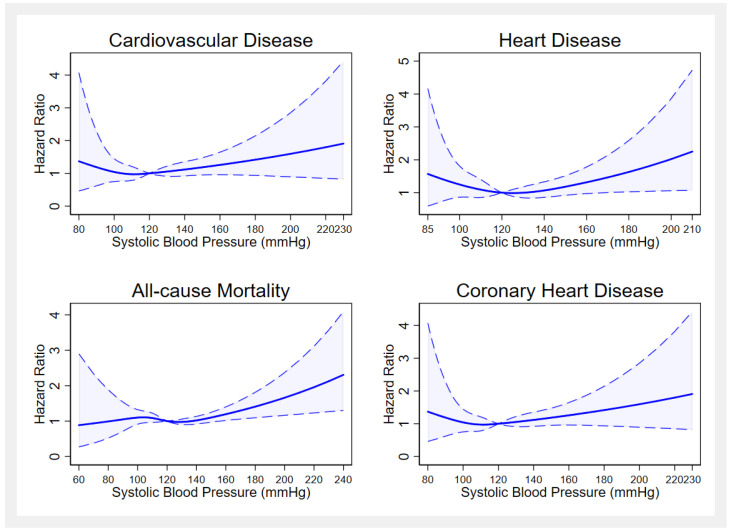
Restricted spline plots of association between systolic blood pressure and mortalities.

**Figure 4 medicina-59-01366-f004:**
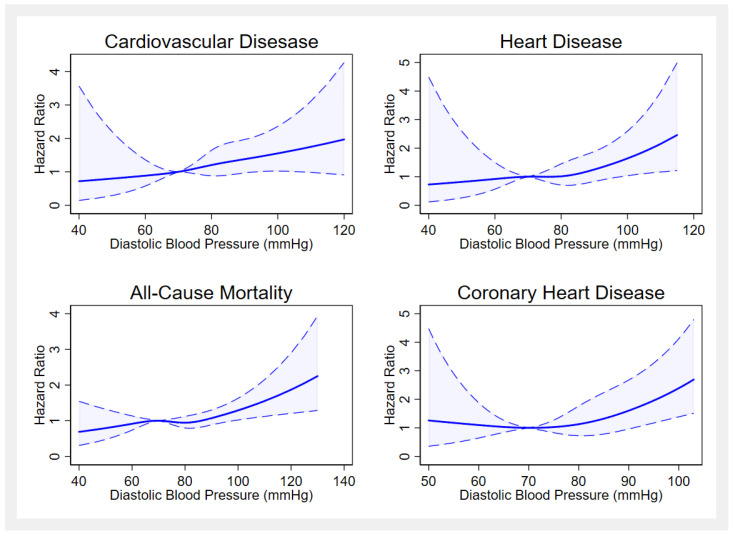
Restricted spline plots of association between diastolic blood pressure and mortalities.

**Figure 5 medicina-59-01366-f005:**
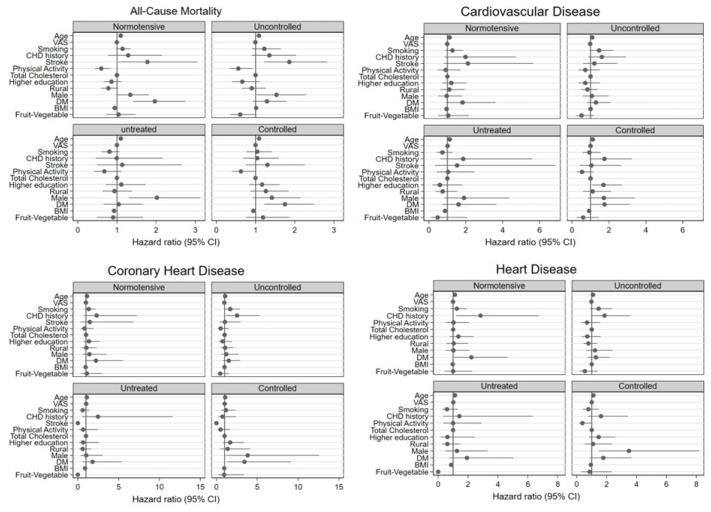
Hazard ratios stratified by mortality and blood pressure status.

**Table 1 medicina-59-01366-t001:** Characteristics of study subjects in Turkey, stratified by blood pressure status.

Variables	Normotensive	Untreated	Treated & Controlled	Treated & Uncontrolled	*p*
**N (%)**	10,921 (75.5)	1484 (10.3)	1066 (7.4)	996 (6.9)	
**Age (years) (%)**					<0.001
18–39	54.0	22.1	4.3	2.8	
40–59	29.9	48.3	44.5	38.5	
≥60	16.1	29.5	51.2	58.7	
**Female (%)**	51.3	47.8	64.9	68.7	<0.001
**Marital status (%)**					
Married	67.7	78.2	76.6	72.5	<0.001
Single/Divorced/widowed	32.3	21.8	23.4	27.5	
**Education status (%)**					
Primary school or less	49.2	71.5	77.1	82.8	<0.001
High school	40.5	21.4	15.0	12.8	
University degree or higher	10.3	7.1	7.9	4.4	
**Fruit and vegetable (%)**	13.6	14.1	18.1	16.7	<0.001
**Smoking (%)**					
Non-smoker	66.3	66.2	73.1	76.7	<0.001
Ex-smoker	7.3	11.9	15.3	13.2	
Current smoker	26.4	21.9	11.6	10.2	
**VAS score**	66.9 ± 30.6	63.84 ± 27.7	59.17 ± 25.33	57.80 ± 24.73	<0.001
**BMI > 30 kg/m^2^ (%)**	17.4	42.2	47.3	56.4	<0.001
**High waist circumference (%)**	26.7	55.5	67.0	75.6	<0.001
**Area lived (%)**					
Rural	30.6	36.2	30.1	38.4	<0.001
**Multimorbidity (%)**					
≥2	5.9	27.5	56.7	59.4	<0.001
**Total Cholesterol (mg/dL)**	174.9 ± 38.9	192.4 ± 40.6	195.1 ± 42.5	199.1 ± 44.2	<0.001
**LDL Cholesterol (mg/dL)**	104.6 ± 34.9	117.9 ± 37.1	118.5 ± 39.3	120.6 ± 37.1	<0.001
**Coronary** **Heart Disease (%)**	1.2	2.9	12.1	17.0	<0.001
**Stroke (%)**	1.3	2.4	6.3	8.1	<0.001
**Diabetes (%)**	5.2	15.9	32.5	35.6	<0.001
**Systolic Blood Pressure (mmHg)**	112.0 ± 11.8	139.0 ± 18.4	119.7 ± 11.0	148.9 ± 17.6	<0.001
**Diastolic Blood Pressure (mmHg)**	70.8 ± 8.8	88.0 ± 11.3	74.4 ± 7.9	87.6 ± 11.8	<0.001
**Framingham Risk Score > 20 (%)**	9.1	22.0	42.7	44.4	<0.001

N, total number; VAS, Visual Analogue Scale; BMI: Body Mass Index.

**Table 2 medicina-59-01366-t002:** Association of hypertension categories with mortalities.

Status	N	Number of Deaths	Number of Deaths Per 100,000 Person-Years (95% CI)	Incidence Rate Ratio (95% CI)	Age- and Sex-Adjusted HR (95% CI)	Multivariable Adjusted HR (95% CI) ^a^
**All-Cause** **Mortality**						
Normal	10,921	266	330.91 (293.95–373.90)	1	1	1
Untreateda	1484	120	1130.62 (948.31–1357.02)	1.15 (0.93–1.41)	1.12 (0.90–1.40)	1.14 (0.92–1.44)
Treated & Controlled	1066	136	1818.09 (1543.14–2155.81)	1.22 (0.99–1.50)	1.24 (1.00–1.54)	1.16 (0.93–1.45)
Treated & Uncontrolled	966	163	2370.35 (2041.49–2766.27)	1.48 (1.22–1.79)	1.44 (1.17–1.78)	1.32 (1.06–1.65)
**Cardiovascular** **Mortality**						
Normal	10,921	62	77.13 (60.50–99.92)	1	1	1
Untreated	1484	35	329.59 (239.15–467.35)	1.25 (0.82–1.91)	1.23 (0.80–1.87)	1.27 (0.83–1.95)
Treated & Controlled	1066	53	708.52 (546.03–936.19)	1.79 (1.23–2.63)	1.73 (1.18–2.53)	1.57 (1.06–2.32)
Treated & Uncontrolled	966	75	1090.652 (875.62–1375.43)	2.42 (1.69–3.48)	2.30 (1.61–3.30)	2.11 (1.46–3.06)
**Heart Disease Mortality**						
Normal	10,921	47	58.46 (44.27–78.89)	1	1	1
Untreated	1484	25	235.42 (161.16–358.10)	1.21 (0.74–1.98)	1.18 (0.72–1.94)	1.21 (0.74–1.99)
Treated & Controlled	1066	35	467.89 (339.76–662.72)	1.61 (1.03–2.55)	1.55 (0.99–2.45)	1.39 (0.88–2.23)
Treated & Uncontrolled	966	58	843.43 (657.26–1100.21)	2.56 (1.69–3.87)	2.43 (1.61–3.68)	2.24 (1.46–3.43)
**Coronary Heart** **Disease Mortality**						
Normal	10,921	31	38.57 (27.41–56.06)	1	1	1
Untreated	1484	20	188.34 (123.34–302.65)	1.21 (0.74–1.98)	1.57 (0.88–2.79)	1.57 (0.87–2.83)
Treated & Controlled	1066	19	253.99 (164.62–413.46)	1.61 (1.02–2.54)	1.43 (0.79–2.59)	1.27 (0.68–2.35)
Treated & Uncontrolled	966	40	581.68 (430.95–804.48)	2.56 (1.70–3.87)	2.92 (1.76–4.86)	2.66 (1.56–4.53)

^a^ Model adjusted for age, gender, education level, area lived(rural), smoking status, fruit and vegetable consumption, physical activity, CHD history, Stroke history, Diabetes Mellitus, Body Mass Index, Total Cholesterol, Visual Analogue Scale.

**Table 3 medicina-59-01366-t003:** Cox regression analyses of the association between blood pressure levels and mortalities.

	Mortality Status *			
	All-Cause Mortality	Cardiovascular Mortality	Heart Disease Mortality	Coronary Heart Disease Mortality
Systolic Blood Pressure				
120<	1.02 (0.83–1.24)	0.94 (0.65–1.35)	0.90 (0.60–1.35)	0.97 (0.59–1.57)
≥120	1.07 (1.02–1.12)	1.06 (0.98–1.14)	1.12 (1.10–1.14)	1.18 (1.06–1.31)
Diastolic Blood Pressure				
<80	1.04 (0.84–1.28)	1.10 (0.73–1.68)	1.05 (0.66–1.71)	0.91 (0.51–1.59)
≥80	1.15 (1.03–1.29)	1.15 (0.96–1.39)	1.31 (1.07–1.61)	1.46 (1.17–1.84)

* Each unit = 10 mmHg.

## Data Availability

Data is available upon request from the Ministry of Health.

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
