# Peer review of "Evaluation of Blood Pressure Status and Mortality in Turkey: Findings from Chronic Diseases and Risk Factors Cohort Study"

_medicina, 2023, doi:10.3390/medicina59081366_

Round 1
Reviewer 1 Report
Interesting perspective epidemiological cohort study to investigate differences in all-cause and cardiovascular related mortality according to HT status by using national data from Turkey. There are though minor concerns regarding the study.
Median follow-up period of 6.2 years should be mention in Abstract.
The introduction need to be shorter.
Please provide the flow chart of study population.
The statistical methods: the multivariable adjusted HR, Which variables were adjusted for this model?
Line 250: Individuals with uncontrolled BP consisted larger proportion of individuals who were female not male. Please correct.
Line 302 to 304: This sentence is vague, please re-write “the hazard ratos increased steeply at higher SBP values, however we did not observe significant relationship between BP and HRs at the lower part of the distribution”
Line 314-320: the result of Table 3 should be completely explain.
Lines 326-330: Please provide significant HR (95% CI) for mortality that mentioned in Figure 4.
The first paragraph of the discussion is not very important and it is better to remove it or make it shorter
Table 1. Please provide abbreviation of VAS score, BMI in footnote
Table 2. Confounding factors that adjusted in multivariable adjusted HR should be mention in footnote.
Figure 2. Title: “restricted spline plots between diastolic blood pressure ….” not systolic.
Author Response
We would like to thank the reviewer for the constructive comments and feedback. We made the necessary changes in the manuscript based on the recommendations of the reviewer.
Comment 1:
Median follow-up period of 6.2 years should be mention in Abstract.
Response 1:
We added the follow-up period to the abstract section
“Median follow-up period was 6.2 years.”
Comment 2:The introduction need to be shorter.
Response 2: We shortened the introduction section.
Comment 3:Please provide the flow chart of study population.
Response 3
The flow chart is constructed and it is now embedded in the manuscript.
Comment 4 The statistical methods: the multivariable adjusted HR, Which variables were adjusted for this model?
Response 4
We adjusted the model using following variables; age, gender, Education level, area lived(rural), smoking status, fruit and vegetable consumption, physical activity, CHD history, Stroke history, Diabetes Mellitus, Body Mass Index, Total Cholesterol, Visual Analogue Scale. We provided this information in both methods section and under Table 2.
Comment 5: Line 250: Individuals with uncontrolled BP consisted larger proportion of individuals who were female not male. Please correct.
Response 5: Thank you for this comment, we corrected this mistake.
Comment 6:Line 302 to 304: This sentence is vague, please re-write “the hazard ratos increased steeply at higher SBP values, however we did not observe significant relationship between BP and HRs at the lower part of the distribution”
Response 6: We rephrased the sentence as follows:
“The hazard ratios for mortality became larger when systolic blood pressure increased over 120mmHg.”
Comment 7:Line 314-320: the result of Table 3 should be completely explain.
Response 7: We added more information regarding analysis used and we explained in more clear way in results section. We used piecewise regression for constructing Table 3. Breakpoints for SBP was 120 mmHg and DBP was 80 mmHg. The data was stratified and two seperate regression model was performed for observations below and above the predefined threshold values. Piecewise regression can be used when independent variable (x) predicts dependent variable(y)differently over ranges of x.
We rephrased the findings regarding Table 3 as follows:
“We used median values of BP as inflection points; 120.0 mmHg for SBP and 80.0 mmHg for DBP were determined as having the lowest hazard ratio for mortalities. The association between blood pressure and risk of mortality was assessed by performing two separate (piecewise) Cox Regression analysis on stratified data based on blood pressure inflection points (Table 3.) Systolic Blood Pressure values higher than 120mmHg was associated with increased risk of all-cause mortality, suggesting that for every 10 mmHg increase in SBP higher than reference value of 120 mmHg was associated with a 7% increased risk in all-cause mortality (HR 1.07, 95% CI: 1.02–1.12). The risk of coronary artery disease related deaths increased by 46% (HR 1.46, 95% CI: 1.17–1.84) for each 10 mmHg increment increase in DBP over 80mmHg.”
Comment 8:Lines 326-330: Please provide significant HR (95% CI) for mortality that mentioned in Figure 4.
Response 8:We corrected the sentence as follows and provided HRs for Diabetes Mellitus.
“We presented HRs regarding covariates stratified by BP groups in Figure 4. The associations of the majority of the independent variables with mortalities were insignificant according to multivariable models. DM was associated with CHD and all-cause mortalities among individuals with controlled BP. Among the individuals with controlled BP, the hazard for mortality in diabetics was significantly higher for all-cause (HR=1.75, 95%CI= 1.22-2.49) and Coronary Heart Disease mortality (HR=3.44, 95%CI=1.29-9.14) compared to individuals without DM.”
Comment 9:The first paragraph of the discussion is not very important and it is better to remove it or make it shorter.
Response 9: We deleted the first paragraph, as this information was already available throughout the manuscript.
Comment 10:Table 1. Please provide abbreviation of VAS score, BMI in footnote
Response 10: We added the abbreviation of VAS score, BMI in footnote in Table 1.
Comment 11:Table 2. Confounding factors that adjusted in multivariable adjusted HR should be mention in footnote.
Response 11: The following information is provided in the footnote: “The model was adjusted for; age, gender, education level, area lived(rural), smoking status, fruit and vegetable consumption, physical activity, CHD history, Stroke history, Diabetes Mellitus, Body Mass Index, Total Cholesterol, Visual Analogue Scale.”
Comment 12: Figure 2. Title: “restricted spline plots between diastolic blood pressure ….” not systolic.
Response 12: We corrected the error we made, thank you.
Reviewer 2 Report
Summary
This study evaluated differences in all-cause and cardiovascular mortality according to HT status by using national data from Turkey. Data from 14,337 individuals who participated in the Chronic Diseases and Risk Factors Survey between the years 2011-2017 were analyzed. Among individuals with HT, 41.8% were untreated, 30.1% were treated and had controlled BP, and 28.1% were treated but had uncontrolled BP. Compared to normotensive subjects, in treated and uncontrolled hypertensive patients the risk of all-cause mortality (HR=1.32, 95%CI= 1.06-1.65), cardiovascular mortality (HR=2.11,95%CI=1.46-3.06), heart disease mortality (HR=2.24, 95%CI=1.46-3.43) and CAD mortality (HR=2.66, 95%CI=1.56-4.53) was significantly higher.
Comments
The abstract could be better reported.
The Introduction section could be shortened and some sentences should be checked for errors.
Page 3, line 120: “2.2. Dependent variables”. I suggest to replace these terms with “Outcomes” or a similar tem.
Page 4, line 170-171: Please, use the same size font.
Page 4, lines 191-193: “Uncontrolled hypertension was defined as having SBP > 140 mmHg or DBP > 90 mmHg in an individual with a history of hypertension despite receiving medication for high blood pressure. Please, report “..SBP >= 140 mmHg or DBP >= 90 mmHg..”
Is FRS extrapolatable to your population?
Table 1: The overall column could be deleted. This column seems unnecessary and deleting it would make the table better readable.
Table 1: In untreated hypertensive patients, mean systolic/diastolic BP were 139/88 mmHg. Please report how many patients had isolated systolic BP, isolated diastolic BP, or both. Moreover, please, report how many patients had mild, moderate or severe hypertension according to 2018 ESC/ESH guidelines. If known, the duration of hypertension in the various groups of hypertensive patients could be an interesting finding to report.
Figure 1 and Table 2: Globally, risk of death appears worse in treated and controlled hypertensive patients than in untreated hypertensive ones. This aspect, at least for me, is surprising. Please, try to explain potential reasons: 1) the younger age of untreated hypertensive patients; 2) the relatively short duration of follow up; 3) a lower duration of hypertension in untreated hypertensive subjects; 4) a higher prevalence of mild hypertension in untreated hypertensive individuals; 5) a lower burden of other major cardiovascular risk factors. If so, your data for the untreated hypertensive population are not representative of a “real untreated hypertensive population”. This aspect should be emphasized in the manuscript.
Page 7, lines 274-275: “Individuals with controlled HT and untreated HT had a lower cumulative probability of mortality compared to other groups for CHD mortality.” This sentence is unclear to me.
Table 2: How many covariates were included in the multivariable Cox regression analysis?
Page 8: “The hazard ratios increased steeply at higher SBP values, however we did not observe significant relationships between BP and HRs at the lower part of the distributions.” Is this statement correct?
Page 9: “Figure 1. Restricted spline plots of association between Systolic Blood Pressure and Mortalities”. This is Figure 2.
Page 9: “Figure 2. Restricted spline plots of association between Systolic Blood Pressure and Mortalities”. This is Figure 3 and regards diastolic BP.
Pages 10-11: “On the other hand, individuals who received treatment and had controlled BP levels had lower mortality risk compared to individuals who were untreated or treated&uncontrolled for HT.” If I am not mistaken, based on data reported in Figure 1 and Table 2, this statement is not correct.
Refs 32 and 33 are the same.
Refs 41 and 43 are the same.
Page 11: “According to the restricted cubic spline regresssion analysis majority of the curves showed J- or U-shaped structure. Especially, higher SBP values resulted with increased risk for mortality and lower values for SBP was associated with increased risk for heart disease related mortalities.” This statement appears in contrast with a previous one.
Globally, the Discussion section should be shortened and focused on main findings.
There are some typos.
English language should be improved.
Please, check reference section.
Moderate editing of English language required
Author Response
Response to Reviewer 2
We would like to thank the reviewer for the constructive comments and feedback. We made the necessary changes in the manuscript based on the recommendations.
Comment 1.The abstract could be better reported.
Response 1:We added the duration of the follow up period and changed the conclusion part.
Comment 2 “The Introduction section could be shortened and some sentences should be checked for errors.”
Response 2: We shortened the introduction section and the text is checked again for errors.
Comment 3 Page 3, line 120: “2.2. Dependent variables”. I suggest to replace these terms with “Outcomes” or a similar term.
Response 3: We replaced the term Dependent variables with Outcomes
Comment 4: Page 4, line 170-171: Please, use the same size font.
Response 4: We checked the font sizes and corrected in the manuscript.
Comment 5 Page 4, lines 191-193: “Uncontrolled hypertension was defined as having SBP > 140 mmHg or DBP > 90 mmHg in an individual with a history of hypertension despite receiving medication for high blood pressure. Please, report “..SBP >= 140 mmHg or DBP >= 90 mmHg..”
Response 5: We made the changes accordingly.
Comment 6: Is FRS extrapolatable to your population?
Response 6: Unfortunately in Turkey there is no country specific and approved Cardiovascular Risk Prediction Equation generated from Turkish data. The main reason for this is the lack of cohort studies. On the other hand some calibrations were made to scoring systems such as SCORE risk (European Society of Cardiology) equation to be used in Turkey. There are several local studies that compare different scoring systems on predicting CVD events or correlation with other diseases. Authors also worked on these scoring equations in several epidemiological and modelling studies. We are aware of the transferability issue hence we added the following information to the limitations section:
“Another limitation is we used Framingham Risk score equation in order to estimate 10 year risk for developing CVD which might resulted with under or overestimation of the risk score values. According to TEKHARF cohort conducted in Turkey, Framingham risk score underestimated the CVD risk[1]. Another research from Turkey found both FRS and SCORE models were reliable for detecting the presence and severity of CVDs[2]. Further research is needed in order to generate risk score equations specific to Turkish settings.”
Comment 7:
Table 1: The overall column could be deleted. This column seems unnecessary and deleting it would make the table better readable.
Response 7: We deleted the overall column in Table 1.
Comment 8: Table 1: In untreated hypertensive patients, mean systolic/diastolic BP were 139/88 mmHg. Please report how many patients had isolated systolic BP, isolated diastolic BP, or both. Moreover, please, report how many patients had mild, moderate or severe hypertension according to 2018 ESC/ESH guidelines. If known, the duration of hypertension in the various groups of hypertensive patients could be an interesting finding to report.
Response 8:Thank you for the comment. We analysed the data as requested and provided the information in results section. Unfortunately the duration of hypertension is not available in the data. We added this information to the limitations section.
“In total 26.5%(n=940) of the individuals with HT had isolated systolic blood pressure. The prevalence of isolated systolic BP among untreated and uncontrolled BP was 46.0% and 25.9%. The prevalence isolated diastolic blood pressure was 9.3%(n=333) among individuals with HT. The prevalence of isolated diastolic BP among untreated and uncontrolled BP was 16.7% and 11.3% respectively. Of the individuals with HT, %57.8 had mild, %9.5 had moderate and 2.6% had severe hypertension. The prevalence of mild, moderate and severe HT among individuals who were untreated was 85.4%, 10.6%, and 4.0% respectively.“
Comment 8:
Figure 1 and Table 2: Globally, risk of death appears worse in treated and controlled hypertensive patients than in untreated hypertensive ones. This aspect, at least for me, is surprising. Please, try to explain potential reasons: 1) the younger age of untreated hypertensive patients; 2) the relatively short duration of follow up; 3) a lower duration of hypertension in untreated hypertensive subjects; 4) a higher prevalence of mild hypertension in untreated hypertensive individuals; 5) a lower burden of other major cardiovascular risk factors. If so, your data for the untreated hypertensive population are not representative of a “real untreated hypertensive population”. This aspect should be emphasized in the manuscript
Response 8: Thank you for this constructive comment. Based on the recommendations we added further details to the discussion section regarding low mortality rate among individuals who were untreated for HT.
“In our study the individuals who had HT but untreated were younger compared to other groups which might explain lower mortality rates in this group.Another reason could be lower burden of other major cardiovascular risk factors as these individuals had lower mean cholesterol levels and they had lower prevalence for DM. Other studies from developing countries had an older population and higher overall risk profile compared to our data. In our study individuals who were untreated also had lower rates for the history of CHD and Stroke and lower rates for presence of multimorbidity. The short duration of follow-up in the study, a lower duration of hypertension in untreated hypertensive subjects due to young age and a higher prevalence of mild hypertension in untreated hypertensive individuals might be other possible explanations.”
Comment 9:
Page 7, lines 274-275: “Individuals with controlled HT and untreated HT had a lower cumulative probability of mortality compared to other groups for CHD mortality.” This sentence is unclear to me.
Response 9: We corrected the sentence as follows.
“Individuals with controlled HT and untreated HT had a lower cumulative probability of CHD mortality compared to individuals with uncontrolled HT.”
Comment 10: Table 2: How many covariates were included in the multivariable Cox regression analysis?
Response 10: We adjusted the model using following variables; age, gender, Education level, area lived(rural), smoking status, fruit and vegetable consumption, physical activity, CHD history, Stroke history, Diabetes Mellitus, Body Mass Index, Total Cholesterol, Visual Analogue Scale. We provided this information in both methods section and under Table 2.
Comment 11. Page 8: “The hazard ratios increased steeply at higher SBP values, however we did not observe significant relationships between BP and HRs at the lower part of the distributions.” Is this statement correct?
We corrected the sentence as follows:
“The hazard ratios for mortality became larger when systolic blood pressure increased over 120mmHg.”
Comment 12. Page 9: “Figure 1. Restricted spline plots of association between Systolic Blood Pressure and Mortalities”. This is Figure 2.
Page 9: “Figure 2. Restricted spline plots of association between Systolic Blood Pressure and Mortalities”. This is Figure 3 and regards diastolic BP.
Response 12:We corrected the titles of the figures.
Comment 13
Pages 10-11: “On the other hand, individuals who received treatment and had controlled BP levels had lower mortality risk compared to individuals who were untreated or treated&uncontrolled for HT.” If I am not mistaken, based on data reported in Figure 1 and Table 2, this statement is not correct.
Response 13. Thank you for the comment,it was a misstatement. We deleted the first paragraph in order to shorten the discussion section, hence this sentence is removed from the manuscript.
Comment 14:
Refs 32 and 33 are the same.
Refs 41 and 43 are the same.
Response 14:Even we used reference software, duplications happened in the references, thank you for this comment, we deleted the duplicates.
Comment 15
Page 11: “According to the restricted cubic spline regresssion analysis majority of the curves showed J- or U-shaped structure. Especially, higher SBP values resulted with increased risk for mortality and lower values for SBP was associated with increased risk for heart disease related mortalities.” This statement appears in contrast with a previous one.
Response 15: Based on reviewers comment we decided to remove this interpretation.
Comment 16: Globally, the Discussion section should be shortened and focused on main findings.
Response 16:
Comment 17:There are some typos.English language should be improved.Please, check reference section.
We checked and corrected typo errors. We rephrased the sentences when necessary. The reference section is corrected, we cleared the duplicates.
- Demirci, D.; Ersan Demirci, D., [Comparison of SCORE-Turkey and SCORE for high-risk countries: A cross-sectional analysis of patients presenting with initial episode of acute coronary syndrome]. Turk Kardiyoloji Dernegi arsivi : Turk Kardiyoloji Derneginin yayin organidir 2019, 47, (8), 646-656.
- Günaydın, Z. Y.; Karagöz, A.; BektaÅŸ, O.; Kaya, A.; Kırış, T.; ErdoÄŸan, G.; Işık, T.; Ayhan, E., Comparison of the Framingham risk and SCORE models in predicting the presence and severity of coronary artery disease considering SYNTAX score. Anatolian journal of cardiology 2016, 16, (6), 412-8.

Reviewer 3 Report
The manuscript conveys the risk for all cause mortality and cardiovascular mortality in a population-based Turkish sample aged 15 years and older over slightly more than 6 years follow-up. BP categories were normotensive, hypertensive untreated, hypertensive controlled, and hypertensive uncontrolled. The observation that higher BP and/or hypertension that is uncontrolled is associated with higher subsequent all-cause and cardiovascular mortality is not novel - except perhaps that little of the data is from Turkish populations. Also, there are some major concerns with the manuscript in its present form:
1) the manuscript is far too long - the introduction should be cut in half as should the discussion
2) in the results section, it is redundant to repeat data verbatim in the text that is in table 1
3) please explicitly state the average or median follow-up of this cohort in the abstract
4)line 130 says that the process of assigning the cause of death as outlined precludes misclassification of the underlying cause of death - this is a gross mis-statement and is not true
5) it is stated that the graphs in figure 2 are J or U-shaped. I am not sure how this was arrived at unless the shape of what appears to be the 95% confidence intervals is being mis-interpreted in this way.
6) lines 386, 387 - the definition of resistant hypertension, which is most accurately called apparent treatment resistant hypertension (because pseudo-resistance cannot be ruled out), is at three antihypertensive drugs of different classes one of which must be a diuretic in a patient with uncontrolled BP or prescription > 3 drugs (one of which is a diuretic) with controlled BP. As written in the text the definition of hypertension is not correct and the subsequent discussion off-base
7) reference 54 which seems to have provided the blueprint for the analyses in this paper, suggests that over the long-term (~19 years) that those with controlled hypertension have similar risk for all-cause and heart-disease deaths - this has not been a universal finding as there is clear residual risk in analyses from the CARDIA study amongst drug-controlled hypertensives compared to normotensives. This should be acknowledged. Further, only deaths are being examined and not morbid cardiovascular events which is another qualifier. In this study, the age- and sex-adjusted HR for all-cause mortality, cardiovascular mortality, and heart disease mortality are higher in those treated and controlled compared to normotensives.
8)a paper by Bhandari and coworkers (Am J Hypertens 2022;35(8):740-744) shed is relevant to the lesser hypertension control in obese patients as this manuscript reported that despite a greater intensity of prescribed antihypertensive drug therapy in lean compared to obese individuals that BP remained higher in the former - implying the need for higher doses to achieve equivalent BP lowering in obese compared to leaner indivudals.
needs minor editing
Author Response
Reviewer 3:
The manuscript conveys the risk for all cause mortality and cardiovascular mortality in a population-based Turkish sample aged 15 years and older over slightly more than 6 years follow-up. BP categories were normotensive, hypertensive untreated, hypertensive controlled, and hypertensive uncontrolled. The observation that higher BP and/or hypertension that is uncontrolled is associated with higher subsequent all-cause and cardiovascular mortality is not novel - except perhaps that little of the data is from Turkish populations. Also, there are some major concerns with the manuscript in its present form:
1) the manuscript is far too long - the introduction should be cut in half as should the discussion
Response 1: Thank you for the comment. We reduced the introduction section and discussion section.
2) in the results section, it is redundant to repeat data verbatim in the text that is in table 1.
Response 2: We removed the part regarding presentation of Framingham risk scores which states numbers in the table.
3) please explicitly state the average or median follow-up of this cohort in the abstract
Response 3: We provided information regarding duration for follow-up period in the abstract section.
“Median follow-up period was 6.2 years”
4)line 130 says that the process of assigning the cause of death as outlined precludes misclassification of the underlying cause of death - this is a gross mis-statement and is not true
Response 4:We removed the following part from the methods section.
“Therefore, the electronic database prevents misclassification in the underlying cause of death.”
5) it is stated that the graphs in figure 2 are J or U-shaped. I am not sure how this was arrived at unless the shape of what appears to be the 95% confidence intervals is being mis-interpreted in this way.
Response 5:Thank you for this comment. We corrected the sentence as follows:
“According to the restricted cubic spline regresssion analysis curves showed non-linear relationship between mortalities and blood pressure levels.”
6) lines 386, 387 - the definition of resistant hypertension, which is most accurately called apparent treatment resistant hypertension (because pseudo-resistance cannot be ruled out), is at three antihypertensive drugs of different classes one of which must be a diuretic in a patient with uncontrolled BP or prescription > 3 drugs (one of which is a diuretic) with controlled BP. As written in the text the definition of hypertension is not correct and the subsequent discussion off-base
Response 6: Thank you for this comment. We removed the whole paragraph about resistant hypertension.
7) reference 54 which seems to have provided the blueprint for the analyses in this paper, suggests that over the long-term (~19 years) that those with controlled hypertension have similar risk for all-cause and heart-disease deaths - this has not been a universal finding as there is clear residual risk in analyses from the CARDIA study amongst drug-controlled hypertensives compared to normotensives. This should be acknowledged. Further, only deaths are being examined and not morbid cardiovascular events which is another qualifier.
Response 7: We added the following explanation to the discussion section.
“However, a study using information from the Multi-Ethnic Study of Atherosclerosis (MESA) and the Coronary Artery Risk Development in Young Adults (CARDIA) study found that people with treated and controlled hypertension had a higher mortality risk than normotensives. That research stated one of the possible explanation as; the participants with controlled blood pressure who are on antihypertensive medication experience significantly higher cumulative BP exposure over time than untreated people. In our research, individuals who were treated and kept under control had higher age- and sex-adjusted HRs for all-cause mortality, cardiovascular mortality, and heart disease mortality compared to normotensives. In addition these individuals are also more likely to experience morbid cardiovascular events compared to untreated and normotensives which was not taken into account in our research.”
Response 7:
8)a paper by Bhandari and coworkers (Am J Hypertens 2022;35(8):740-744) shed is relevant to the lesser hypertension control in obese patients as this manuscript reported that despite a greater intensity of prescribed antihypertensive drug therapy in lean compared to obese individuals that BP remained higher in the former - implying the need for higher doses to achieve equivalent BP lowering in obese compared to leaner indivudals.
Response 8: We added the following information to the discussion section
“A recent study found that; obese people's blood pressure remained higher despite receiving more antihypertensive medication than leaner individuals, suggesting that obese people would require higher dosages to drop their blood pressure to the same extent as leaner people[47].

Round 2
Reviewer 2 Report
The authors answered the questions and the manuscript has improved.
Page 5, lines 176-179: Please, check.
Page 7, lines 289-293: Please, replace “isolated SBP” with “isolated systolic hypertension”, and “isolated DBP” with “isolated diastolic hypertension”.
Page 15, lines 515-519: “In our research, individuals who were treated and kept under control had higher age- and sex-adjusted HRs for all-cause mortality, cardiovascular mor-tality, and heart disease mortality compared to normotensives. In addition, these individ-uals are also more likely to experience morbid cardiovascular events compared to un-treated and normotensives which was not taken into account in our research.” The last sentence is unclear to me.
Minor editing of English language required
Author Response
Dear editor,
We would like to thank to the reviewers for their support and constructive comments. We made the necessary revisions in the manuscript as requested.
Reviewer 2:
Comment 1: Page 5, lines 176-179: Please, check.
Response 1:We corrected the error for definition of rural areas.
“According to the Turkish Statistical Institution's (TURKSTAT) classification, settlements with a population of at least 20,000 people were classified as urban, while those with a population below 20,000 people were accepted as rural areas”
Comment 2: Page 7, lines 289-293: Please, replace “isolated SBP” with “isolated systolic hypertension”, and “isolated DBP” with “isolated diastolic hypertension”.
Response 2:We made the necessary changes in the text.
Comment3: Page 15, lines 515-519: “In our research, individuals who were treated and kept under control had higher age- and sex-adjusted HRs for all-cause mortality, cardiovascular mor-tality, and heart disease mortality compared to normotensives. In addition, these individ-uals are also more likely to experience morbid cardiovascular events compared to un-treated and normotensives which was not taken into account in our research.” The last sentence is unclear to me.
Thank you for the comment. We removed the last sentence to avoid misunderstanding. We rephrased the last sentence and added it to limitations part.
“In our study, only mortality were evaluated; however, the occurrence of morbid cardiovascular events may vary by HT status, which can be considered in future research.”